# Effect of Cultivar, Plant Spacing and Harvesting Age on Yield, Characteristics, Chemical Composition, and Anthocyanin Composition of Purple Napier Grass

**DOI:** 10.3390/ani13010010

**Published:** 2022-12-20

**Authors:** Narawich Onjai-uea, Siwaporn Paengkoum, Nittaya Taethaisong, Sorasak Thongpea, Boontum Sinpru, Jariya Surakhunthod, Weerada Meethip, Rayudika Aprilia Patindra Purba, Pramote Paengkoum

**Affiliations:** 1School of Animal Technology and Innovation, Institute of Agricultural Technology, Suranaree University of Technology, Muang, Nakhon Ratchasima 30000, Thailand; 2Program in Agriculture, Faculty of Science and Technology, Nakhon Ratchasima Rajabhat University, Muang, Nakhon Ratchasima 30000, Thailand

**Keywords:** purple Napier grass (*Pennisetum purpureum* “Prince”), Napier Pakchong 1 (*Pennisetum purpureum × Pennisetum americanum* cv. Pakchong 1), crude protein, anthocyanin composition, morphological characteristics

## Abstract

**Simple Summary:**

Napier grass is a vegetative propagation and can survive repeated cuttings and rapidly regrow. It is favorable for leafy cattle. Purple Napier grass (*Pennisetum purpureum* “Prince”), as a dwarf Napier grass, contains anthocyanin content. Anthocyanins are pigments found in many plants. Anthocyanin was found to have several functional and biological properties, including antioxidant activity. We found that variations in cultivars, plant spacing, and harvesting age are crucial to increasing their performance and are the main factors affecting their characteristics, chemical composition, and anthocyanin composition.

**Abstract:**

Purple Napier grass is a semi-dwarf, purple-leaved Napier grass. The purple color is anthocyanins. Anthocyanin is classified as a group of flavonoids. It has antioxidant properties. The objective of this study was to determine the effect of plant spacing and harvesting age on the forage yield, morphological characteristics, chemical composition, and anthocyanin composition of purple Napier grass. An experiment was conducted to determine the effect of plant spacing and harvesting age on the forage yield, morphological characteristics, chemical composition, and anthocyanin composition of purple Napier grass when grown on a sandy soil. The cultivars were Napier Pakchong 1 (*Pennisetum purpureum* × *Pennisetum americanum* cv. Pakchong 1) and purple Napier grass (*Pennisetum purpureum* “Prince”), with plant spacings of 50 × 50, 50 × 75, and 75 × 75 cm, and the harvesting ages were 45, 60, and 75 days. The experiment was a 2 × 3 × 3 factorial layout in a randomized complete block design with four replications, for a total of 72 plots, each 5 × 5 m. The purple Napier grass had a higher number of tillers per plant than the Napier Pakchong 1 grass. The LSR value (leaf/stem ratio) was influenced by the interaction of cultivar × plant spacing × harvesting age. The purple Napier grass planted at 75 × 75 cm for 45 days had the highest LSR value. The crude protein of the purple Napier grass, the grass planted at 75 × 75 cm, and the grass for 45 days were significantly higher than the other treatments. The purple Napier grass planted at 75 × 75 cm for 45 days had the highest (*p* < 0.05) anthocyanin content. It was concluded that purple Napier grass planted at 75 × 75 cm for 45 days would contain the proper number of tillers per plant, LSR value, chemical composition for ruminants, and the highest anthocyanin composition.

## 1. Introduction

Napier grass is the principal forage in humid tropical countries, with the capability to develop high dry matter per unit area, especially in comparison to several other grass crops [1]. Napier grass has already been introduced in all tropical countries and is widely cultivated throughout Southeast Asia, where the average rainfall is an estimated 1000 mm [1]. Napier grass can grow in soil where many plant species are critical but do not grow well [2]. Napier grass is a vegetative propagation and can survive repeated cuttings and rapidly regrow, producing a product that is favorable for leafy cattle [3].

Napier Pakchong 1 (*Pennisetum purpureum* × *Pennisetum americanum* cv. Pakchong 1) is a tall grass cultivar. Napier Pakchong 1 is a cross between ordinary Napier (*Pennisetum purpureum*) and pearl millet (*Pennisetum americanum*), developed by Krailas Kiyothong, an animal nutritionist and plant breeder at the Department of Livestock Development in Pak Chong, Nakhon Ratchasima Province, in northeastern Thailand. Purple Napier grass is a semi-dwarf, purple-foliaged Napier grass (*Pennisetum purpureum* “Prince”), jointly released by the United States Department of Agriculture (USDA) and the University of Georgia College of Agricultural and Environmental Sciences (UGA). Purple Napier grass was imported by the Dairy Farming Promotion Organization of Thailand (DPO), which is located in the Muak Lek District of Saraburi Province, Thailand, at the study site and stored until transplantation.

Variations in cultivars cause differences in the morphological characteristics. Grasses with higher growth rates usually have lower nutrient quality because the relatively high DM yield enhances the need to form more structural carbohydrates [4], while grasses with a greater growth performance actually have a higher nutrient quality [5]. The precise timing of defoliation to achieve an optimum yield and quality is generally related to the age of the plant, and the provided recommendations vary. A grass defoliation interval of 70 days or when the plant reaches 120–150 cm is recommended for optimal results and forage quality [6]. Moreover, Ferreira [7] suggests grass defoliation at 60–85 days of age, while Moran [8] noticed that the grass can be harvested after 25–30 days in the rainy season or 50–60 days in the dry season. These studies found that increasing the plant’s growing age increased the dry matter yield while decreasing the quality.

Anthocyanin has been discovered to have a variety of biological and functional properties, including antioxidant activity [9]. The use of bioactive compound-containing forages is a natural and viable alternative for encouraging a beneficial antioxidant balance, improving animal performance. In addition, studies on the morphological characteristics of the Napier grass cultivar on yield and quality have been widely applied [10], while there has been little study on the purple Napier grass cultivar; thus, the effect of the plant spacing and harvesting age of purple Napier grass on the forage yield, morphological characteristics, chemical composition, and anthocyanin composition remains unclear. Therefore, the aim of this study was to investigate the effect of plant spacing and harvesting age of purple Napier grass on the forage yield, morphological characteristics, chemical composition, and anthocyanin composition.

## 2. Materials and Methods

### 2.1. Experiment Site

The experiment was conducted at Suranaree University of Technology (SUT) goat farm, Nakhon Ratchasima, Thailand (latitude: 14°52′48″ N, longitude: 102°00′15″ E; elevation: 222 m; temperature: 30 °C; rainfall: 1112 mm), from June 2016 to June 2017. The soil of the experimental site has been classified into the Korat soil series (sandy loam, Oxic Paleustults) [11]. To plant the Napier grass, old clumps were divided into rooted slips, each of which had two tillers 10 cm long in addition to the roots. These slips were planted by making a small hole in the ground with a hand hoe. The experiment had three factors: the first was two grass cultivars Napier Pakchong 1 (*Pennisetum purpureum* × *Pennisetum americanum* cv. Pakchong 1) and purple Napier grass (*Pennisetum purpureum* “Prince”); the second was three spaces 50 × 50, 50 × 75, and 75 × 75 cm; and the third factor was three harvesting ages 45, 60, and 75 days, which were planted in experimental plots (5 × 5 m) and harvested after regrowth cutting. Approximately 312.5 kg/ha of NPK 15-15-15 fertilizer and approximately 62.5 kg/ha of NPK 46-0-0 fertilizer were applied to all grass plots before planting and after each cutting interval, respectively. Sprinkler irrigation was used every 5 days or as needed to ensure optimal soil moisture conditions for pasture growth [12].

### 2.2. Data Collection and Sampling

The morphological characteristic measurements included the plant height, number of tillers per plant, and leaf/stem ratio (LSR). The plant height, number of tillers per plant, and LSR were measured at the time of cutting. Before each harvest, the plant height was measured from its base to where the last leaf on the stem emerged using a meter rule on five randomly selected culms per plot [13]. The grass was cut close to the soil surface and the first cuttings were on day 120 after planting. The grass was collected eight times every 45 days, six times every 60 days, and five times every 75 days.

### 2.3. Yield and Chemical Composition

The forage yields for each cutting interval were measured using the quadrat technique (size: 50 × 50 cm) and converted to kilograms per hectare (kg/ha). Each sample was divided into two parts. The first part was dried for 48 h in a hot-air oven at 105 °C to determine its dry matter (DM) content. The forages were dried in a hot-air oven at 60 °C for 72 h, and then ground to pass through a 1 mm^2^ mesh screen and analyzed for their chemical compositions in the second part of the sample used for the nutritive value analysis. Total yield (DM/ha) was calculated using dry weight of collected forage samples. Total N was determined using the Kjeldahl method, and the crude protein (CP) was calculated by multiplying the N content by 6.25, following a previous observation [14]. The ether extract (EE) and ash content were quantified by AOAC [15]. The neutral detergent fiber (NDF) and acid detergent fiber (ADF) were estimated using the methods described by Van Soest et al. [16].

### 2.4. Anthocyanin Composition Analysis

The anthocyanin composition, including cyanidin-3-glucoside (C3G), pelargonidin-3-glucoside (P3G), delphinidin (Del), peonidin-3-O-glucoside (Peo3G), malvidin-3-O-glucoside (M3G), cyanidin (CYA), pelargonidin (Pel), malvidin (Mal), and total anthocyanin, was determined using the modified HPLC method [17]. Previous methods [2,18] were used to adapt the standard stock solution, calibration standard, and sample preparation for quality control. Grass samples were adjusted to pH 4 with 1% hydrochloric acid and pretreated with an acetone/chloroform liquid-liquid extraction (70:30, *v/v*), then centrifuged at 10,000 r/min at 4 °C for 15 min, incubated at room temperature for 4 h, and the supernatant was collected for anthocyanin composition. The analysis of the specimen was performed with an HPLC and diode array detector (DAD). The anthocyanin content extraction was achieved on the column C18 Symmetry (mobile phase: A, acetonitrile (CH_3_CN); B, 10% acetic acid/5% CH_3_CN/1% phosphoric acid in deionized water). The time period was 30 min, followed by a delay of 5 min before the next injection. Other conditions included a sample temperature of 4 °C, an injection volume of 20 L, a flow rate of 0.8 mL/min, a column temperature of 25 °C, and a DAD wavelength of 520 nm.

### 2.5. Statistical Analysis

All the statistical calculations were analyzed using the general linear model (GLM) procedure of Statistical Analysis System 9.4 [19] according to a 2 × 3 × 3 factorial in a randomized complete block design with four replications of each treatment arranged within three blocks according to the area slope. Significant differences (*p* < 0.05) among treatment combinations were determined using the Tukey–Kramer test according to Steel and Torrie [20]. The statistical model for the analysis of the data was:Y_ijk_ = µ + C_i_ + S_j_ + A_k_ + C_i_ × S_j_ + C_i_ × A_k_ + S_j_ × A_k_ + C_i_ × S_j_ × A_k_ + b_lk_ +ε_ijkl_
where:

Y_ijk_ = All dependent variables;

µ = The overall mean;

C_i_ = the effect of the ith cultivar of Napier Pakchong 1 (NP-1) and purple Napier (PN);

S_j_ = The effect of the jth plant spacing of Napier grass (50 × 50, 50 × 75, and 75 × 75 cm);

A_k_ = The effect of the kth harvesting age of Napier grass (45, 60, and 75 days);

C_i_ × S_j_ = The interaction of cultivar × plant spacing;

C_i_ × A_k_ = The interaction of cultivar × harvesting age;

S_j_ × A_k_ = The interaction of plant spacing × harvesting age;

C_i_ × S_j_ × A_k_ = The interaction of cultivar × plant spacing × harvesting age;

Blk = The effect of the block in the environment (area slope);

ε_ijkl_ = Residual.

Comparisons of least square means were performed using the Tukey–Kramer test. For all the data analyzed, *p* values equal to or less than 0.05 were considered statistically significant.

## 3. Results

### 3.1. Morphological Characteristics, Chemical Composition, and Yield

The leaf/stem ratio (LSR) value of the Napier grass was influenced by cultivar × space × age (*p* < 0.05) (Table 1) in the purple Napier grass planted at 75 × 75 cm with a harvesting age of 45 days and had the highest LSR value. The plant height was significantly influenced by the interaction of space × age (*p* < 0.05) (Table 2). The plant spacing of 50 × 50 cm with a harvesting age of 75 days had the highest plant height value. The plant height parameter of the Napier grass was influenced by cultivar, space, and age (*p* < 0.05) (Table 3). The Napier Pakchong 1 grass had a plant height value higher than the purple Napier grass. The plant spacing of 50 × 50 cm had the highest plant height value. The harvesting age of 75 days had the highest plant height value. The number of tillers per plant had significant effects on cultivar, space, and age (*p* < 0.05) (Table 3). The purple Napier grass had a higher number of tillers per plant than the Napier Pakchong 1 grass. The plant spacing of 75 × 75 cm had the highest number of tillers per plant value. The harvesting age of 60 days had the highest number of tillers per plant value.

The ash of the Napier grass was influenced by cultivar × space × age (*p* < 0.05) (Table 1) in the Napier Pakchong 1 planted at 75 × 75 cm with a harvesting age of 75 days and had the highest ash value. The results for the dry matter (DM), crude protein (CP), ether extract (EE), neutral detergent fiber (NDF), hemicellulose, and lignin were significantly influenced by the interaction of space × age (*p* < 0.05) (Table 2). The plant spacing of 75 × 75 cm with the harvesting age of 45 days had the highest percentage of CP and EE. The plant spacing of 75 × 75 cm with the harvesting age of 75 days had the highest percentage of DM, NDF, hemicellulose, and lignin.

The CP, crude fiber (CF), ADF (acid detergent fiber), and cellulose were significantly affected by the main effect of cultivar, space, and age (Table 3). The purple Napier grass was significantly higher in CP content than the Napier Pakchong 1 grass. The Napier Pakchong 1 grass was significantly higher in ADF and cellulose contents than the purple Napier grass. The plant spacing of 75 × 75 cm was significantly higher in CP and CF contents than for 50 × 50 and 50 × 75 cm. The harvesting age of 45 days was significantly higher in CP content than for 60 and 75 days. The harvesting age of 75 days was significantly higher in CF, ADF, and cellulose contents than for 45 and 60 days. The dry matter yield of the Napier grass was significantly influenced by the interaction of cultivar × space × age (*p* < 0.05) (Figure 1) for the dry matter yield of the two Napier cultivars, including Napier Pakchong 1 and purple Napier. The highest DM yield of the Napier Pakchong 1 grass was planted at 75 × 75 cm with a harvesting age of 75 days.

### 3.2. Anthocyanin Composition

The results of the anthocyanin composition included cyanidin-3-glucoside (C3G), pelargonidin-3-glucoside (P3G), delphinidin (Del), peonidin-3-O-glucoside (Peo3G), malvidin-3-O-glucoside (M3G), cyanidin (CYA), pelargonidin (Pel), malvidin (Mal), and total anthocyanin. The interaction of cultivar × space × age (*p* < 0.05) had a significant effect on the composition of P3G and M3G (Table 1). The purple Napier grass planted at 75 × 75 cm and harvested at 45 days had the highest P3G and M3G contents.

The interaction of cultivar × space, cultivar × age, and space × age (*p* < 0.05) (Table 2) had a significant effect on the composition of C3G, Del, Peo3G, CYA, Pel, Mal, and total anthocyanin. The purple Napier grass planted at 75 × 75 cm had the highest C3G, Del, Peo3G, CYA, Pel, Mal, and total anthocyanin contents. The purple Napier grass with a harvesting age of 45 days had the highest C3G, Del, Peo3G, CYA, Pel, Mal, and total anthocyanin contents. The plant spacing of 75 × 75 cm with the harvesting age of 45 days had the highest percentage of C3G, Peo3G, Pel, Mal, and total anthocyanin.

## 4. Discussion

### 4.1. Morphological Characteristics, Chemical Composition, and Yield

Cultivar, plant spacing, and harvesting age had a significant impact on the DM yield of the Napier grass. These factors would affect photosynthetic activity, higher DM yield, and plant height as well as more tillers and leaves per plant and a higher chemical composition value [21].

The Napier Pakchong 1 and purple Napier grass were divided into tall grass cultivars and short grass cultivars (dwarf Napier). Because of the faster growth rate of Napier Pakchong 1, which accumulates more dry matter than purple Napier grass, the average DM yield, plant height, CF, NDF, ADF, cellulose, and lignin was higher for the Napier Pakchong 1 grass than the purple Napier grass [21,22]. Furthermore, the plant height of Napier Pakchong 1 was greater than that of purple Napier, resulting in a higher DM yield. There was a positive correlation between the plant height and DM yield of Napier Pakchong 1 (tall grass cultivar), and the significantly larger tillers in the purple Napier highlight the vigorous growth and adaptation to the hot and humid climate [21,23,24].

The short cultivars had a leafier structure than the tall cultivars, which contributed to their higher nutritive value [25]. The purple Napier (dwarf grass) had a higher LSR than the Napier Pakchong 1 (tall grass). The leaf/stem ratio (LSR) is one of the criteria used to evaluate pasture grass quality, because a higher proportion of leaves to stem indicates a higher nutritive value. Optimal spacing is required for Napier grass to grow, optimal grass yield for ruminant, and be of high quality. Although a narrow spacing was more productive, a wider spacing resulted in higher DM yields, tillers per plant, LSRs, and better chemical compositions. Individual plants in the Napier grass grew taller with longer internodes and slender, thin, and weak stalks as a result of low light exposure and, thus, low photosynthetic output [26].

Because of the wider spacing, light could easily penetrate to the base of the plant, which may have stimulated tiller development, resulting in higher DM yields, LSRs, and chemical compositions [27]. Individual plants can support more tillers because competition for nutrients is reduced when plants are spaced further apart [27]. When sufficient space is available, the ability of Napier grass to increase the number of tillers per plant increases, with variation due to the variable nutritional areas and light access [26].

Lower tiller counts at a close plant spacing may be due to the intense competition for resources, such as light, space, and nutrients [26,28]. Because of the increased competition for light, growth and tillering capacity are reduced. Interplant competition in Napier grass causes rapid and exhausting height increments, resulting in neighboring plants producing weak tillers because of overcrowding [26,28]. As a result, instead of spreading laterally by bearing more tillers, competitor plants are forced to grow upright in order to dominate other tillers produced on the same plant [28]. If the density is kept above the optimum, the overall demand for resources increases, resulting in plant stress [29].

The higher DM, plant height, fiber content (ADF and NDF), CF, and lignin content were related to the harvesting day, while the number of tillers per plant, LSR, CP, and EE content decreased. The high DM yield, plant height, fiber compounds (ADF and NDF), CF, and lignin content were harvested at a later harvesting age due to the growing maturity of the Napier grass, resulting in weight gain from biomass production [30,31].

At 75 days, the number of tillers per plant increased. The defoliation and activation of basal buds may lead to an increase in the number of tillers per plant, thereby removing apical dominance [32]. Because increasing the number of tillers per plant is probably an adaptive feature to tolerate frequent defoliation by re-establishing a lost photosynthetic area and maintaining the basal area, the number of tillers per plant was lower in the long harvesting age, resulting in the higher mortality of the tillers under reduced cutting frequency [33]. A high tiller production per plant indicates not only stable productivity but also improved persistence of adverse environmental conditions [34,35,36]. Tiller production is an important factor in grassland resistance to deterioration due to the fact of aging [37].

The LSR of the Napier grass decreased as the harvesting age increased. The leaf fraction decreased significantly from 45 to 75 days. The LSR of the Napier Pakchong 1 grass was lower than the purple Napier grass, indicating that it had a higher proportion of stems than leaves. As a result, rather than purple Napier grass, the Napier Pakchong 1 grass can be classified as a high-yielding and stemmy cultivar. The purple Napier grass was the leafier cultivar (high LSR value), which has an impact on the nutritive quality because the leaves contain more nutrients and are less fibrous than the stem fraction [38].

The percentage of CP at 60 and 75 days after planting was lower than at 45 days after planting. This causes different stages of maturity. The plant changed from a vegetative phase to a reproductive phase. The reproductive stage of the greater proportion of stems caused increased fiber content, but the crude protein content decreased. The quality will be reduced due to the increased maturity [39]. Budiman [40] reported that the crude protein content of the Napier grass King and Mott cultivars decreased with increasing defoliation intervals. Adjei and Fianu [1] reported that the average CP content in the leaves and stems of forage legumes decreased from 22.5 to 17.5% and from 11.9 to 9.4% and that the corresponding rise in the average CF content in the leaves and stems was from 20.0 to 26.8% and 27.1 to 31.9% with a longer harvesting age (60, 90, and 120 days).

A harvest at 45 days resulted in significantly lower dry matter yields. While the protein content was high, it was insufficient to compensate for the significantly reduced forage production [41,42]. For a hot humid tropical climate, a 45-day harvesting age appears to be ideal. The results show a high LSR value and an acceptable percentage of CP at 45 days at harvest, which supports this theory. The nutrient value of the Napier grass harvested at a young age in this study was excellent, with a high percentage of CP, a limiting nutrient in tropical forages.

Most of the Napier grass harvested at 60 and 75 days, however, had a CP content well above 7%, which is the level below which ruminant voluntary intake may be depressed [14]. All of the forage produced would provide enough energy and protein to sustain some level of production above that required for maintenance [14]. Despite some reduction in the CP content and an increase in the NDF content, the grass allowed to grow until 75 days of age at harvest produced a much higher DM yield without a significant reduction in quality. In any pasture situation, compromises between quality and yield must be made when deciding at what stage to harvest or graze a crop or pasture [42].

In the current study, the Napier Pakchong 1 had a high NDF and a low CP content, whereas the purple Napier grass had a low NDF and a high CP content. The Napier Pakchong 1 grass produced more DM than the purple Napier grass. As a result, two very different types of Napier grass would be chosen for ruminants, but the balance of the DM yield and chemical composition would be considered the best possible forage properties for ruminants in the current study.

### 4.2. Anthocyanin Composition

Cultivars, plant spacing, and harvesting age all had a significant impact on the anthocyanin composition of the Napier grass. The purple Napier grass had a higher anthocyanin content when grown with a wider plant spacing and harvested at an earlier age. The purple Napier grass had a higher anthocyanin composition than the Napier Pakchong 1 grass, because the anthocyanin content in the red leaves and stems was significantly higher than that in the green leaves and stems, and the highest anthocyanin content was detected in the red leaves, with the leaves having a higher anthocyanin content than the stems [43].

Chlorophylls, carotenoids, and flavonoids are among the pigments that determine leaf color, which is an important external indicator of plant quality [6,44]. The high expression of chalcone synthase (CHS) helps in the production of anthocyanins and other flavonoids. However, the CHS expression was significantly lower in the green leaves of the Napier grass compared to the purple and light purple leaves [43].

The purple Napier grass was a leafy grass with a heavier leaf fraction than the stem fraction and a high leaf/stem ratio (LSR) due to the leaf structure of the short cultivars [25]. The leaves (high LSR) had a higher proportion of anthocyanins than the stem, resulting in a positive anthocyanin composition.

Phenylalanine and tyrosine are key players in the production of specialized phenylpropanoids, a large and diverse family of phenylpropanoids. Plants use phenylpropanoids to adapt to changing environmental conditions and to protect themselves from pathogens [45]. Both amino acids are the sources of phenylpropanoids, which are primarily associated with specialized metabolic pathways and are products of the shikimate and aromatic amino acid pathways. The flavonoid and anthocyanin pathways, as well as the benzenoid, stilbenoid, and lignin pathways, are all part of the phenylpropanoid pathway [46,47,48,49,50].

The different phenylpropanoid pathways resulting from phenylalanine and tyrosine catabolism are differentially induced, depending on the plant, developmental, and environmental conditions, as well as biotic and abiotic stresses [51,52,53,54]. The overproduction of phenylalanine and tyrosine had no effect on the total anthocyanin concentrations [55]. The longer harvest ages stimulated earlier flowering and fruit development, so the assimilates were distributed across many sinks. This partitioning might also have decreased the total nitrogen content and protein synthesis in the leaves, as a longer harvesting age might have led to earlier ripening, decreased biological properties, and reduced overall nitrogen and lignification in the leaves, resulting in decreased protein synthesis [19,56,57].

Furthermore, sunlight and high temperatures boost anthocyanin synthesis [4,58,59]. The most important external factor influencing anthocyanin synthesis is solar radiation [60,61,62]. The red color of the skin in many plants is caused by several flavonoid genes required for anthocyanin synthesis being transcribed in a coordinated manner in response to light exposure [43]. Anthocyanin synthesis is expected to increase in response to high temperature stress, which may increase the amount of ROS in mitochondria via respiration. High temperatures can cause an increase in respiratory rates and ethylene production. It was assumed that higher temperatures would also increase gene expression and anthocyanin production [59].

## 5. Conclusions

The purple Napier grass planted at 75 × 75 cm with a harvesting day of 45 days would be suitable for ruminants in terms of morphology (the highest tillers per plant and LSR value), chemical composition (the highest CP content), and anthocyanin composition. Further in vitro and in vivo studies should be performed for greater consistency and clarity.

## Figures and Tables

**Figure 1 animals-13-00010-f001:**
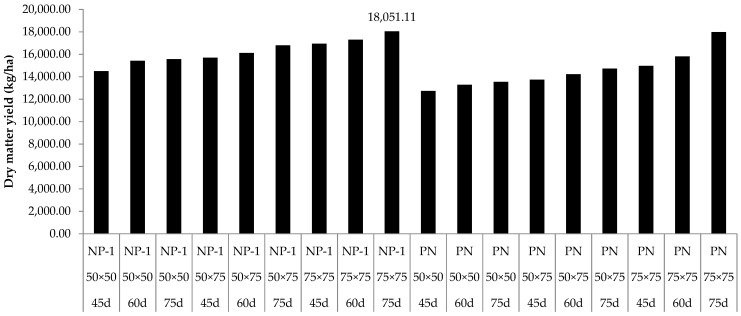
Dry matter yield (kg/ha) of fresh Napier grass as affected by cultivar × plant spacing (cm^2^) × harvesting age (days). Cultivar = the cultivar of Napier grass; space = plant spacing (cm^2^); age = harvesting age (days); NP-1 = Napier Pakchong 1 grass; PN = purple Napier grass.

**Table 1 animals-13-00010-t001:** The effect of the interaction of cultivar × plant spacing × harvesting age on the parameters of fresh Napier grass.

Interaction			LSR ^2^	Ash	Anthocyanin Composition	
Cultivar ^1^	Space	Age			P3G ^3^	M3G ^4^
NP-1	50 × 50	45	1.72	9.41	0.08	0.06
		60	1.44	9.76	0.04	0.03
		75	0.96	10.08	0.01	0.01
	50 × 75	45	1.78	10.81	0.10	0.07
		60	1.51	11.20	0.07	0.04
		75	0.99	12.57	0.01	0.02
	75 × 75	45	1.94	12.93	0.18	0.10
		60	1.64	13.65	0.07	0.04
		75	1.53	15.44	0.02	0.03
PN	50 × 50	45	1.74	6.03	0.20	0.14
		60	1.45	6.75	0.12	0.08
		75	0.98	6.95	0.02	0.03
	50 × 75	45	1.79	7.09	0.26	0.19
		60	1.59	8.03	0.17	0.10
		75	1.03	8.69	0.04	0.05
	75 × 75	45	2.00	8.85	0.46	0.27
		60	1.67	8.91	0.18	0.10
		75	1.58	9.05	0.06	0.07
		SEM	0.044	0.345	0.013	0.008
*p*-Value						
Cultivar × Space × Age			0.043	0.0001	0.0001	0.0004

Means followed by a different letter within the same column are significantly different (*p* < 0.05). SEM = standard error of the mean. ^1^ Cultivar = the cultivar of Napier grass; space = plant spacing (cm^2^); age = harvesting age (days); NP-1 = Napier Pakchong 1 grass; PN = purple Napier grass. ^2^ LSR = leaf/stem ratio. ^3^ P3G = pelargonidin-3-glucoside. ^4^ M3G = malvidin-3-O-glucoside.

**Table 2 animals-13-00010-t002:** The effect of the interaction of cultivar × plant spacing, cultivar × harvesting age, and plant spacing × harvesting age on plant height, chemical composition, and anthocyanin content of fresh Napier grass.

Interaction		Plant Height (cm)	Chemical Composition (%)						Anthocyanin Content (mg/g Dry Weight)						
			DM ^2^	CP	EE	NDF	Hemi	Lignin	C3G ^3^	Del	Peo3G	CYA	Pel	Mal	Total Ant.
Cultivar × Space ^1^															
NP-1	50 × 50	156.07	21.37	7.84	2.93	73.50	25.57	2.52	0.24	0.04	0.08	0.05	0.04	0.22	0.75
	50 × 75	147.67	21.60	8.23	3.37	73.99	28.54	3.28	0.31	0.05	0.12	0.06	0.06	0.23	0.93
	75 × 75	129.00	22.32	9.05	3.37	76.37	29.82	3.46	0.42	0.06	0.15	0.07	0.07	0.27	1.16
PN	50 × 50	141.11	21.18	7.90	2.90	72.19	26.29	2.60	0.61	0.11	0.19	0.12	0.11	0.54	1.88
	50 × 75	136.22	21.36	8.73	3.24	73.36	25.09	3.31	0.78	0.13	0.30	0.14	0.16	0.56	2.33
	75 × 75	116.22	22.19	9.25	3.31	74.08	29.37	3.34	1.06	0.16	0.39	0.17	0.18	0.66	2.90
Cultivar × Age															
NP-1	45	131.45	20.23	9.87	3.54	64.20	18.82	2.73	0.76	0.08	0.19	0.09	0.09	0.33	1.71
	60	147.67	21.60	8.23	3.25	73.99	28.88	3.28	0.31	0.05	0.12	0.06	0.06	0.23	0.93
	75	163.68	22.96	4.09	2.75	76.37	29.82	3.69	0.09	0.02	0.05	0.04	0.03	0.20	0.45
PN	45	115.70	20.09	9.95	3.45	62.55	19.75	2.63	1.90	0.20	0.50	0.24	0.24	0.82	4.34
	60	136.22	21.36	8.73	3.24	73.70	27.64	3.31	0.78	0.13	0.30	0.14	0.16	0.56	2.33
	75	160.32	23.64	4.28	2.62	73.36	26.29	3.33	0.24	0.06	0.12	0.09	0.08	0.51	1.18
Space × Age															
50 × 50	45	123.58	19.76	9.58	3.40	62.49	19.29	2.31	1.07	0.13	0.30	0.15	0.14	0.50	2.52
	60	148.59	21.28	7.87	2.92	73.05	28.26	2.56	0.43	0.08	0.14	0.09	0.08	0.38	1.32
	75	183.65	22.93	3.92	2.32	73.07	25.93	3.22	0.05	0.04	0.04	0.06	0.05	0.33	0.57
50 × 75	45	126.22	20.16	9.91	3.50	64.12	20.38	2.70	1.33	0.14	0.35	0.17	0.17	0.58	3.03
	60	141.95	21.48	8.48	3.25	73.85	27.76	3.30	0.55	0.09	0.21	0.10	0.11	0.40	1.63
	75	162.00	22.98	4.19	3.00	74.87	27.46	3.51	0.17	0.04	0.09	0.07	0.06	0.36	0.82
75 × 75	45	115.05	21.60	11.72	3.60	63.38	18.68	2.76	1.56	0.17	0.45	0.20	0.18	0.68	3.73
	60	122.61	22.26	9.15	3.34	75.23	29.60	3.40	0.74	0.11	0.27	0.12	0.13	0.47	2.03
	75	161.00	24.51	7.51	2.74	82.19	33.16	3.57	0.37	0.07	0.13	0.08	0.06	0.38	1.16
SEM		4.071	0.202	0.335	0.058	0.897	0.706	0.059	0.074	0.008	0.020	0.009	0.008	0.028	0.162
*p*-Value															
Cultivar × Space		0.499	0.111	0.169	0.078	0.802	0.398	0.392	0.0001	0.0003	0.0001	0.013	0.010	0.005	0.0001
Cultivar × Age		0.652	0.233	0.744	0.417	0.955	0.470	0.965	0.0001	0.0001	0.0001	0.0001	0.0001	0.0001	0.0001
Space × Age		0.011	0.0001	0.0001	0.0001	0.0100	0.0001	0.0001	0.005	0.943	0.0009	0.360	0.005	0.004	0.0007

Means followed by a different letter within the same column are significantly different (*p* < 0.05). SEM = standard error of the mean. ^1^ Cultivar = the cultivar of Napier grass; space = plant spacing (cm^2^); age = harvesting age (days); NP-1 = Napier Pakchong 1 grass; PN = purple Napier grass. ^2^ DM = dry matter; CP = crude protein; EE = ether extract; NDF = neutral detergent fiber; Hemi = hemicellulose. ^3^ C3G = cyanidin-3-glucoside; Del = delphinidin; Peo3G = peonidin-3-O-glucoside; Cya = cyanidin; Pel = pelargonidin; Mal = malvidin; Total Ant. = Total anthocyanin.

**Table 3 animals-13-00010-t003:** The effect of cultivar, plant spacing (cm^2^), and harvesting age (days) on plant height, the number of tillers/plant, and chemical composition of fresh Napier grass.

Main Effect	Plant Height (cm)	Number of Tillers/Plant	Chemical Composition (% DM)			
			CP ^2^	CF	ADF	Cellulose
Cultivar ^1^						
NP-1	148.51	27.99	8.23	36.58	46.22	43.15
PN	136.96	31.08	8.73	35.59	45.34	42.34
Space						
50 × 50	151.94	25.39	7.12	34.92	45.15	42.45
50 × 75	143.39	30.90	7.53	36.24	45.75	42.58
75 × 75	132.89	32.30	9.46	37.10	46.45	43.21
Age						
45	121.61	20.26	10.40	33.59	43.98	41.40
60	137.72	32.31	8.50	37.06	45.50	42.36
75	168.88	36.03	5.20	37.61	47.86	44.49
SEM	4.071	1.097	0.335	0.306	0.326	0.294
*p*-Value						
Cultivar	<0.0001	0.0004	<0.0001	0.187	0.01	0.01
Space	<0.0001	0.0001	<0.0001	0.0001	0.05	0.247
Age	<0.0001	0.0001	<0.0001	0.0001	0.0001	0.0001

Means followed by a different letter within the same column are significantly different (*p* < 0.05). SEM = standard error of mean. ^1^ Cultivar = the cultivar of Napier grass; space = plant spacing (cm^2^); age = harvesting age (days); NP-1 = Napier Pakchong 1 grass; PN = purple Napier grass. ^2^ CF = crude fiber; ADF = acid detergent fiber.

## Data Availability

All data are contained within the article.

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
