# Peer review of "Effect of Cultivar, Plant Spacing and Harvesting Age on Yield, Characteristics, Chemical Composition, and Anthocyanin Composition of Purple Napier Grass"

_animals, 2022, doi:10.3390/ani13010010_

Round 1

Reviewer 1 Report

The manuscript reports results of a study on purple Napier grass characteristics (morphology, yield, and chemical composition) as affected by plant spacing and harvesting age, comparing it to a Napier grass cultivar. The argument is interesting and deserving to be explored. However, some aspects need to be better described: it is not clearly explained the difference between Napier grass and purple Napier grass (add latin names), and the trial design and the sampling method are not clear.

The manuscript need to be revised by a native English speaker. It is poorly written and somewhere difficult to understand. The results should be focused on significant interaction only when these are significant. As a rule: results should be shown for the highest interaction level (3 way interaction) if this is significant. if not, the 2 way interaction should be shown. In case these are also not significant, the main effect should be shown. Following this rule, tables 1, 2, 3, and 4 should be deleted and results reported in tables and figures according to the significant interactions. The discussion section need to be deepen improved. The arguments are messy and often the sentences are not well linked to the following. It often is unclear where the authors are stating their results or results from a cited reference. Sometimes, cited studies need to be better contextualize reporting information on them. Furthermore, authors should make order in the discussion. The discussions are organized in parts discussing the analysed factors, and this is a good starting point. However, whithin the part, the authors should put together discussion about parameters.

The abstract should also be improved.

Author Response

Response to Reviewer 1 Comments

Point 1: The manuscript reports results of a study on purple Napier grass characteristics (morphology, yield, and chemical composition) as affected by plant spacing and harvesting age, comparing it to a Napier grass cultivar. The argument is interesting and deserving to be explored. However, some aspects need to be better described: it is not clearly explained the difference between Napier grass and purple Napier grass (add latin names), and the trial design and the sampling method are not clear.

Response 1: Please see the attachment.

Point 2: The manuscript need to be revised by a native English speaker. It is poorly written and somewhere difficult to understand. The results should be focused on significant interaction only when these are significant. As a rule: results should be shown for the highest interaction level (3 way interaction) if this is significant. if not, the 2 way interaction should be shown. In case these are also not significant, the main effect should be shown. Following this rule, tables 1, 2, 3, and 4 should be deleted and results reported in tables and figures according to the significant interactions. The discussion section need to be deepen improved. The arguments are messy and often the sentences are not well linked to the following. It often is unclear where the authors are stating their results or results from a cited reference. Sometimes, cited studies need to be better contextualize reporting information on them. Furthermore, authors should make order in the discussion. The discussions are organized in parts discussing the analysed factors, and this is a good starting point. However, whithin the part, the authors should put together discussion about parameters.

Response 2: Please see the attachment.

Point 3: The abstract should also be improved.

Response 3: Please see the attachment.

Point 4: Title: the title is misleading. Here only purple Napier grass is reported, so cultivar cannot have an effect on its characteristics and chemicals. I suggest to simplify the title.

Response 4: Please see the attachment.

Point 5: Keywords: avoid words already reported in the title.

Response 5: Please see the attachment.

Point 6: line 26: Could you briefly describe the trial?

Response 6: An experiment was conducted to determine the effect of plant spacing and harvesting age on the forage yield, morphological characteristics, chemical composition, and anthocyanin composition of purple Napier grass when grown on a sandy soil. The cultivars were Napier Pakchong 1 (Pennisetum purpureum cv. Pakchong 1) and purple Napier grass (Pennisetum purpureum “Prince”), with plant spacings of 50 × 50, 50 × 75, and 75 × 75 cm2 , and the harvesting ages were 45, 60, and 75 days. The experiment was a 2 × 3 × 3 factorial layout in a randomized complete block design with four replications, for a total of 72 plots, each 5 × 5 m2.

Point 7: line 29: “CP”, avoid acronymus here.

Response 7: Please see the attachment.

Point 8: line 39: provide latin name (or names) of Napier grass

Response 8: Please see the attachment.

Point 9: lines 39-40: need reference.

Response 9: Please see the attachment.

Point 10: lines 51-52: need reference.

Response 10: Please see the attachment.

Point 11: lines 52-55: not clear, rephrase.

Response 11: Please see the attachment.

Point 12: line 54: this term “results” is misleading. What do you mean? Yield?

Response 12: According to Walaiphan [57], for optimal yield and forage quality, a grass defoliation interval of 70 days or when the plant reaches 120-150 cm is recommended.

Point 13: line 72: do you mean the study was conducted from June 2016 to June 2017? Or in June 206 and in June 2017? Two growing seasons?

Response 13: The study was conducted from June 2016 to June 2017.

Point 14: lines 73-76: please, rewrite the text in a more scientific way.

Response 14: Please see the attachment.

Point 15: lines 86-95: this is an important aspect. move it to the introduction

Response 15: Please see the attachment.

Point 16: line 97: How did you measure plant height?

Response 16: Before each harvest, the plant height was measured from its base to where the last leaf on the stem emerged using a meter rule on five randomly selected culms per plot.

Point 17: lines 103-105: this is not in line with the following section. Please, explain better.

Response 17: Please see the attachment.

Point 18: lines 105-106: rephrase in a more scientific way.

Response 18: Please see the attachment.

Point 19: lines 106-107: “according….[64].” Move it in the following section.

Response 19: Please see the attachment.

Point 20: line 108: rephrase as "Yield and chemical composition".

Response 20: Please see the attachment.

Point 21: lines 109-113: not clear which are the samples and the sub-sample. Are the subsamples composed by leaves or stems? Are they dried at both 60 and 105°C? Do you calculate and analyse the total yield expressed in DM? In the results only the whole samples seem to be reported.

Response 21: The forage yields for each cutting interval were measured at 50 × 50 cm2 and then hand-clipped and weighed. Each sample (whole grass) was divided into two parts. The first part was dried for 48 hours in a hot-air oven at 105 °C to determine its dry matter (DM) content. The forages were dried in a hot-air oven at 60 °C for 72 hours, and then ground to pass through a 1 mm2 mesh screen and analyzed for their chemical compositions in the second part of the sample used for the nutritive value analysis.

Point 22: lines 134-135: the design description is unclear: how many replicates did you perform? In the agricultural field experiments blocks are used to include field condition variability in the analysis, so usually randomized block design are used. Furthermore, split-plot design is often needed to apply treatments, as it is difficult to manage the trial in the field as a completed randomized design. Could you better describe your experimental design?

Response 22: All the statistical calculations were analyzed using the general linear model (GLM) procedure of Statistical Analysis System 9.4 according to a 2 × 3 × 3 factorial in a randomized complete block design with four replications of each treatment arranged within three blocks according to the area slope. The main plot treatments are measured with less precision than they are in a randomized complete block design. When missing data occur, the analysis is more complex than for a randomized complete block design with missing data. Different treatment comparisons have different basic error variances which make the analysis more complex than with the randomized complete block design, especially if some unusual type of comparison is being made.

Point 23: Line 138-140: perhaps this sentences should be moved at the end of the section.

Response 23: Please see the attachment.

Point 24: Line 153: results: as a rule, results should be shown for the highest interaction level (3 way interaction) if this is significant. If not, the 2 way interaction should be shown. In case these are also not significant, the main effect should be shown. Tables are difficult for the reader and gives too much and unnecessary information when the 3 way interaction is not significant. Please, make tables or graphs for the significant interactions only.

Response 24: Please see the attachment.

Point 25: Line 155: data of yield are not reported in table 1.

Response 25: Please see the attachment.

Point 26: Lines 159-160, 166-167, 168-169, 171-172, 173-174, 181-182, 184-185, 187, 189-191, 192-193, 194-195, 196-197, 199-200, 209, 215: delete the sentences, they are a repetition of previous statements.

Response 26: Please see the attachment.

Point 27: Lines 163,164, 167, 168, 170, 172, 189, 198, 201, 205, 206, 207, 208, 210, 211, 212, 213, 214, 216, 217, 218, 219, 234, 236, 240, 241, 244, 245, 246, 252, 253, 257, 258, 263, 264, 265: please, report in brackets the mean values.

Response 27: Please see the attachment.

Point 28: Line 176: if “(2.00)” is the LSR, move it at the end of the sentence.

Response 28: Please see the attachment.

Point 29: Line 179: for Table 2 see comment to table 1.

Response 29: Please see the attachment.

Point 30: Line 180: delete “respectively”.

Response 30: Please see the attachment.

Point 31: Line 185: delete “The interaction….when”.

Response 31: Please see the attachment.

Point 32: Lines 185-186: at which age, as the 3 way interaction is significant?

Response 32: The ash of the Napier grass was influenced by cultivar × space × age (p < 0.05) in the Napier Pakchong 1 planted at 75 × 75 cm2 with a harvesting age of 75 days and had the highest ash value.

Point 33: Lines 188, 194, 205, 207, 210, 212, 216, 218, 233: delete “(p<0.05)”.

Response 33: Please see the attachment.

Point 34: lines 195-196: which cultivar, as the 3 way interaction was significant?

Response 34: The interaction of space × age influenced the NDF content significantly (p < 0.05). The plant spacing of 75 × 75 cm2 and harvested at 75 days had the highest percentage of NDF.

Point 35: Line 198, 200: specify which cultivar instead of “the grass”.

Response 35: Please see the attachment.

Point 36: Lines 220-224: I suggest to use only DM yield for this manuscript. As the other parameters has not been described in the results, they could be skip out. Thay don't give more information than % in DM.

Response 36: Please see the attachment.

Point 37: Lines 226-228: The description of analysed anthocyanin should be also added in the m&m section.

Response 37: Please see the attachment.

Point 38: Lines 229-230: the 3 way interaction is not significant for all parameters. Table 3: see comment on tables 1 and 2.

Response 38: Please see the attachment.

Point 39: Lines 231, 232, 239, 243: the cited interaction is not clear.

Response 39: Please see the attachment.

Point 40: Line 232: change “This” with “The following”.

Response 40: Please see the attachment.

Point 41: line 233: the highest?

Response 41: Please see the attachment.

Point 42: line 236: which age, as the 3 way interaction was significant?

Response 42: The purple Napier grass planted at 75 × 75 cm2 and harvested at 45 days had the highest P3G and M3G contents.

Point 43: Line 237-238: delete “, which….(p<0.05)”.

Response 43: Please see the attachment.

Point 44: Lines 261-262: delete “, respectively…..age”.

Response 44: Please see the attachment.

Point 45: Line 266: delete “respectively”.

Response 45: Please see the attachment.

Point 46: Line 267: delete “,while…..no effect,”.

Response 46: Please see the attachment.

Point 47: Lines 304-305: this seems not a stand-alone sentence.

Response 47: Please see the attachment.

Point 48: Line 305: at which characteristics is “these” referred to?

Response 48: Please see the attachment.

Point 49: Lines 305-307: not clear.

Response 49: Please see the attachment.

Point 50: Lines 308-312: not clear if this comes from your results or from reference. Furthermore, the distinction on dwarf and tall should be introduced in the introduction.

Response 50: Please see the attachment.

Point 51: Lines 317-318: results about leaves and stems fractions are not reported in the results section.

Response 51: Please see the attachment.

Point 52: Lines 320-322, 323-325: these sentences seem out of context.

Response 52: Please see the attachment.

Point 53: Line 329: you should specify that that this come from your results.

Response 53: Please see the attachment.

Point 54: Lines 330-337: rephrase. The text is difficult to read.

Response 54: Please see the attachment.

Point 55: Lines 338-344, 345-355: rephrase. The text is difficult to read. Furthermore, it is unclear where the authors are stating their results or results from a cited reference.

Response 55: Please see the attachment.

Point 56: Lines 356-358: rephrase.

Response 56: Please see the attachment.

Point 57: Line 364: rephrase.

Response 57: Please see the attachment.

Point 58: Lines 399-400, 400-401: need reference.

Response 58: Please see the attachment.

Point 59: Lines 401-403: not clear. When you cite a study, you should report the main information of the study and you should link it to your results.

Response 59: Please see the attachment.

Reviewer 2 Report

Observations and considerations can be found in the attached document.

Author Response

Response to Reviewer 2 Comments

Page 2

Point 1: Considering that the evaluation of anthocyanin composition is one of the main differentials in the study, it is necessary to present more elements in the introduction that justify its relevance for animal production.

Response 1: Please see the attachment.

Point 2: Inform the period of the experiment. Inform the climatic data during the period, mainly precipitation and temperature.

Response 2: Please see the attachment.

Page 3

Point 1: The samples were collected in how many grass cutting cycles? Or was it just a cut click? Kindness informs in the material and methods.

Response 1: The grass was collected eight times every 45 days, six times every 60 days, and five times every 75 days.

Point 2: Inform in case of interaction, the procedure for unfolding and analysis.

Response 2: The experiment had three factors: the first was two grass cultivars Napier Pakchong 1 (Pennisetum purpureum cv. Pakchong 1) and purple Napier grass (Pennisetum purpureum “Prince”); the second was three spaces 50 × 50, 50 × 75, and 75 × 75 cm2; and the third factor was three harvesting ages 45, 60, and 75 days.

Point 3: Considering that it was an experiment carried out in the field and it may be influenced by the conditions of the terrain, inform the reason for not having been in randomized blocks.

Response 3: Please see the attachment.

Page 4

Point 1: Considering that the effects of significance are described in the results text, the presentation of the tables below becomes repetitive. It is recommended to replace the respective tables with tables and graphs of the significant effects studied with the respective unfolding of the interactions, as observed below.

Response 1: Please see the attachment.

Point 2: In table 1 there is no productivity. Productivity values are in Table 4.

Response 2: Please see the attachment.

Point 3: Information regarding the data in Table 4.

Response 3: Please see the attachment.

Point 4: You are repeating the previous information.

Response 4: Please see the attachment.

Point 5: It is necessary to carry out the unfolding of the interaction to study the averages referring to spacing as a function of harvest age and also to study the averages of harvest age as a function of spacing. It is recommended the representation in the form of a table with the respective significance, as well as the difference identified by letters. Bearing in mind that the Tukey Test will be used to compare averages.

Response 5: Please see the attachment.

Point 6: In this case, it is recommended to represent and compare the general means of NP-1 (148.51 cm) with PN (136.96 cm).

Response 6: Please see the attachment.

Point 7: Observe the recommendation for the same condition above.

Here it is necessary to consider the following situation, the spacing factor:

50x50=> Average 25.39

50x75=> Average 30.90

75x75=> Average 31.83

As reported, the 75x75 spacing showed the highest value of tillers per plant. As for the 50x75 spacing situation, how is it compared to 50x50 and 75x75 spacing? It is recommended the representation and comparison of the general averages by graphs and/or tables with the respective identification of the differences by letters, when performing the Tukey test.

Recommendation valid for age.

Response 7: Please see the attachment.

Point 8: It is recommended to carry out the unfolding of the triple interaction, so that a detailed comparison study of the effects of the factors and objective conclusions will be possible.

Response 8: Please see the attachment.

Point 9: Observe the recommendation for the same condition above..

Response 9: Please see the attachment.

Point 10: Realce

Response 10: Please see the attachment.

Page 5

Point 1-6: Observe the recommendation for the same condition above. Regarding the data in table 4, there was a triple interaction for all variables, which requires unfolding for a better understanding and discussion of the data.

Response 1-6: Please see the attachment.

Page 6

Point 1-3: Observe the recommendation for the same condition above.

Response 1-3: Please see the attachment.

Page 7

Point 1-22: Realce

Response 1-22: Please see the attachment.

Page 9

Point 1-21: Realce

Response 1-21: Please see the attachment.

Page 10

Point 1-11: Pay attention to possible changes in the discussion due to the unfolding of interactions in the results

Response 1-11: Please see the attachment.

Page 11

Point 1: They are plants with different growth habits, Rhodes grass has smaller tillers, allowing the plant to invest in the number of tillers, compared to Napier, which has large tillers, requiring greater investment. Therefore, a reference to a forage with the same growth pattern would be coherent.

Response 1: Please see the attachment.

Point 2: Plants harvested at 75 days were larger because they had more time to grow, a difference of 15 and 30 days when compared to plants harvested at 60 and 45 days, respectively. I believe that the reference to photoperiod needs to be better contextualized, since it refers to day length. Considering that the period during which the experiment was carried out was not informed in the Material and methods, it is believed that it is unlikely that drastic changes in the photoperiod have occurred in the space of 15 to 30 days.

Response 2: Please see the attachment.

Page 13

Point 1: What is the relationship between Crude Protein and Anthocyanin?

Response 1: Phenylalanine and tyrosine are key players in the production of specialized phenylpropanoids, a large and diverse family of phenylpropanoids. Plants use phenylpropanoids to adapt to changing environmental conditions and to protect themselves from pathogens. Both amino acids are the sources of phenylpropanoids, which are primarily associated with specialized metabolic pathways and are products of the shikimate and aromatic amino acid pathways. The flavonoid and anthocyanin pathways, as well as the benzenoid, stilbenoid, and lignin pathways, are all part of the phenylpropanoid pathway.

Point 2: The information needs to be better contextualized to be more consistent with the results. Considering that the effects of nitrogen and potassium were not evaluated, as well as there are no evaluations on senescence.

Response 2: Please see the attachment.

Page 14

Point 1: After unfolding the interactions and updating the discussion, verify that the conclusion is maintained.

Response 1: Please see the attachment.

Reviewer 3 Report

The description of the Introduction is not rich enough, and the summary of the experiment is not detailed enough. It is suggested to enrich the content

In Part 2.1, whether the application of chemical fertilizer twice has an impact on the experiment results, and how to exclude its effect? How is the dose determined?

The discussion part is too jumbled, and it is suggested to focus on the key points for in-depth analysis and highlight important research results

Author Response

Response to Reviewer 3 Comments

Point 1: The abstract should be concise.

Response 1: Please see the attachment.

Point 2: The description of the Introduction is not rich enough, and the summary of the experiment is not detailed enough. It is suggested to enrich the content

Response 2: Please see the attachment.

Point 3: In Part 2.1, whether the application of chemical fertilizer twice has an impact on the experiment results, and how to exclude its effect? How is the dose determined?

Response 3: The grass was applied with chemical fertilizer in this experiment. There was no effect on the experiment because commercial fertilizers were used in this experiment. Therefore, the quality and quantity of fertilizer are absolutely accurate.

Point 4: The discussion part is too jumbled, and it is suggested to focus on the key points for in-depth analysis and highlight important research results

Response 4: Please see the attachment.

Point 5: The description of the results is too complicated. The figures in parentheses are repetitive expressions of the contents in the table. Is it necessary.

Response 5: Please see the attachment.

Round 2

Reviewer 1 Report

The authors have deeply improved the manuscript. However, some parts still need to be better described and some comments has not been addressed exhaustively. In the next revision I ask to the authors to answer to each comment directly in the word file: this make the process easier for the reviewer.

The Latin name of species/cultivars involved in the study should be check, for example in the introduction Napier Pakchong 1 is reported to be a cross between two species at line 55, but the Latin name at line 54 reports only one species.

The results section is the most confused. It is difficult to review this section, as it is not clearly stated the significant interactions. Please, provide a table with significance of the effects for all parameters. It seems that authors not reported results only for significant interactions, but also for not significant interactions. Furthermore, some data are reported two times (for example for DM the interaction between space and age is reported in the table 2 and figure 2).

The main issue is that the study has been conducted for one year only, that is considered not significant in describing plant characteristics due to climate seasonal variation.

Specific comments:

Lines 31, 33, 114, 121: Delete “2” after “cm”.

Line 53 and so on: the references are not numbered correctly.

Line 66: the distinction between “high and dwarf” need reference.

Lines 115-117: if fresh samples has been divided in leaf and stems, how could you divide the samples with whole grass, as describe in lines 122?

Lines 121-122: is Total yield calculated using fresh or dried weight? change with "total yield (DM/ha) was calculated using dry weight of collected forage samples." and move it after the next sentence.

Line 122: delete “whole grass”.

Lines 124-126: Should this be the same of method described at lines 115-117? Whit the same samples?

Results: please provide a table with results of the statistical analysis, this is a starting point to order results.

Discussion: I would like to review this section when the result section will be corrected.

Author Response

Response to Reviewer 1 Comments

Point 1: The Latin name of species/cultivars involved in the study should be check, for example in the introduction Napier Pakchong 1 is reported to be a cross between two species at line 55, but the Latin name at line 54 reports only one species.

Response 1: Pennisetum purpureum × Pennisetum americanum cv. Pakchong 1.

Point 2: The results section is the most confused. It is difficult to review this section, as it is not clearly stated the significant interactions. Please, provide a table with significance of the effects for all parameters. It seems that authors not reported results only for significant interactions, but also for not significant interactions. Furthermore, some data are reported two times (for example for DM the interaction between space and age is reported in the table 2 and figure 2).

Response 2: Please see the attachment.

Point 3: The main issue is that the study has been conducted for one year only, that is considered not significant in describing plant characteristics due to climate seasonal variation.

Response 3: Plant characteristics must be reported because they are related to their nutritional value when used as ruminant feed.

Point 4: Lines 31, 33, 114, 121: Delete “2” after “cm”.

Response 4: Please see the attachment.

Point 5: Line 53 and so on: the references are not numbered correctly.

Response 5: Please see the attachment.

Point 6: Line 66: the distinction between “high and dwarf” need reference.

Response 6: Please see the attachment.

Point 7: Lines 115-117: if fresh samples has been divided in leaf and stems, how could you divide the samples with whole grass, as describe in lines 122?

Response 7: Please see the attachment.

Point 8: Lines 121-122: is Total yield calculated using fresh or dried weight? change with "total yield (DM/ha) was calculated using dry weight of collected forage samples." and move it after the next sentence.

Response 8: dried weight and please see the attachment.

Point 9: Line 122: delete “whole grass”.

Response 9: Please see the attachment.

Point 10: Lines 124-126: Should this be the same of method described at lines 115-117? Whit the same samples?

Response 10: Please see the attachment.

Point 11: Results: please provide a table with results of the statistical analysis, this is a starting point to order results. Discussion: I would like to review this section when the result section will be corrected.

Response 11: Please see the attachment.

Reviewer 2 Report

Comments directed to the editor.

Round 3

Reviewer 1 Report

The result section is now clearer and the discussions are appropriated.

Specific comments:

Lines 22 and 48: change “Napier grass (Pennisetum purpureum)” with a more general term including both purple napier grass and Napier Pakchong 1.

The reference has been numbered in a wrong way: “References must be numbered in order of appearance in the text” (from author’s instructions).

Materials and methods: please, add information about temperatures and rainfall of the study period.

Lines 248-251, 267-269: is this a result coming from your study or from the Reference?

Lines 244-245: I’m not sure CF, NDF, ADF, cellulose, and lignin contents depend on growth speed.

Line 265: the term “yield” is not clear.

Line 267: add “better” before “chemicals”.

Line 249: change “lead to an increase” with “affect”.

In the last revision I’ve not find an answer to my comment:

Lines 124-126: Should this be the same of method described at lines 115-117? Whit the same samples?

I suggest to consider my suggestion to publish this manuscript as Short communication due to the lack of repetition in time of the experiment. I think this is very important for maintaining scientific relevance of studies on agriculture.

Author Response

Response to Reviewer 1 Comments

Specific comments:

Point 1: Lines 22 and 48: change “Napier grass (Pennisetum purpureum)” with a more general term including both purple napier grass and Napier Pakchong 1.

Response 1: Please see the attachment.

Point 2: The reference has been numbered in a wrong way: “References must be numbered in order of appearance in the text” (from author’s instructions).

Response 2: Please see the attachment.

Point 3: Materials and methods: please, add information about temperatures and rainfall of the study period.

Response 3: The experiment was conducted at Suranaree University of Technology (SUT) goat farm, Nakhon Ratchasima, Thailand (latitude: 14°52’48”N, longitude: 102°00’15”E; elevation: 222 m; temperature: 30 °C; rainfall: 1112mm ), from June 2016 to June 2017.

Point 4: Lines 248-251, 267-269: is this a result coming from your study or from the Reference?

Response 4: Lines 248-251: this a result coming from the reference (Cultivar, plant spacing, and harvesting age had a significant impact on the DM yield of the Napier grass. These factors would affect photosynthetic activity, higher DM yield, and plant height as well as more tillers and leaves per plant and a higher chemical composition value [50].). Line 267-269: this a result coming from the reference (Individual plants in the Napier grass grew taller with longer internodes and slender, thin, and weak stalks as a result of low light exposure and, thus, low photosynthetic output [62].).

Point 5: Lines 244-245: I’m not sure CF, NDF, ADF, cellulose, and lignin contents depend on growth speed.

Response 5: The CF, NDF, ADF, cellulose, and lignin contents depend on cultivar, plant spacing, and harvesting age were the objectives of this experiment.

Point 6: Line 265: the term “yield” is not clear.

Response 6: Optimal spacing is required for Napier grass to grow, optimal grass yield for ruminant, and be of high quality.

Point 7: Line 267: add “better” before “chemicals”.

Response 7: Please see the attachment.

Point 8: Line 249: change “lead to an increase” with “affect”.

Response 8: Please see the attachment.

Point 9: Lines 124-126: Should this be the same of method described at lines 115-117? Whit the same samples?

Response 9: Lines 124-126 are the same method described in lines 115-117, which is the same samples.

Round 4

Reviewer 1 Report

The manuscript can be published in the present form.